# Investigating the distribution of antibiotic resistance genes in relation to bacterial, fungal, and functional diversity in a hay field

Carolina Oliveira de Santana,[1,2] Pieter Spealman,[3] Conrad Vispo,[4] David Gresham,[3] Sage Saccomanno,[5] Christopher N. LaFratta,[5] Swapan S. Jain,[5] Robert S. Dungan,[6] Gabriel G. Perron[2,3]

**ABSTRACT** The spread of antibiotic resistance in pathogenic bacteria is one of the most pressing public health threats. While recent work has shown the importance of environmental reservoirs in the emergence of antibiotic resistance genes (ARGs), it is unclear which features of microbial diversity relate to ARGs of clinical relevance. Here, we perform a small-scale study of the relationships between bacterial, fungal, and functional diversity with the distribution of two classes of ARGs (clinical and environmental) along a single transect located in an aging hay field on an otherwise active farm. This transect spans a length of several hundred meters, increasing in distance from an agricultural access road and stream. We use *16S rRNA* and *ITS* amplicon sequencing to measure bacterial and fungal diversity, respectively, in combination with whole-genome sequencing to characterize functional and ARG diversity. We find increasing bacterial and functional diversity along the transect, as well as distinct community structures for both bacteria and fungi. While we find that the diversity of environmental ARGs is significantly correlated with both bacterial and fungal diversity, clinical ARG diversity significantly decreased as fungal diversity increased. Our results suggest that while bacterial diversity increases with distance from the road and stream, this diversity is correlated with the diversity of the environmental ARGs, this trend is not observed for ARGs of clinical relevance, which appear to be largely driven by the variety of fungal groups in the environment.

**IMPORTANCE** Antibiotic resistance is often studied in hospitals and clinical settings, but much less is known about how resistance genes are distributed in everyday environments, such as agricultural soils. Hay fields are widespread, actively managed ecosystems that sit at the interface of natural microbial communities, farming practices, and food systems. In this study, we examine how antibiotic resistance genes are associated with broader patterns of bacterial diversity, fungal communities, and soil functional potential in a working hay field. By integrating resistance gene profiles with ecological measures of microbial diversity, we move beyond simply cataloging resistance and instead place it within a community and ecosystem context. This approach helps clarify whether resistance genes are linked to specific microbial groups, overall biodiversity, or functional traits related to soil processes. Our findings provide insight into how antibiotic resistance persists and is structured in low-input agricultural systems, contributing to a more complete understanding of environmental reservoirs of resistance and informing discussions about sustainable land management and public health risk.

**KEYWORDS** antibiotic resistance, microbial communities, fungal-bacterial interactions

Address correspondence to Gabriel G. Perron, gperron@bard.edu.

The authors declare a conflict of interest.

The spread of antibiotic resistance in pathogenic bacteria is one of the most pressing public health threats of the 21st century (1, 2). Current models state that roughly 5 million deaths can be associated with antibiotic-resistant bacteria globally and that this number could double by 2050 if no actions are taken (3, 4). Antibiotic pollution resulting from medical, industrial, domestic, and agricultural activities (5–7) combined with the release of antibiotic-resistant bacteria of public health concerns via wastewater and farming (8–10) creates selective environments on an unprecedented global scale (11–13). Importantly, while recent work has revealed that novel antibiotic-resistant bacteria strains often emerge from environmental reservoirs (14–17), regulators rarely pay attention to the spread of antibiotic-resistance genes (ARGs) in the environment (16, 18).

Agricultural soil ecosystems are also increasingly recognized as an important reservoir of antibiotic-resistant bacteria (10, 19) as most soils harbor a vast diversity of ARGs (20, 21). Indeed, due to the presence of antibiotic-producing bacteria like *Streptomyces* spp., or fungi like *Penicillium*, many ARGs are uniquely present in soils (22–24) and pre-date antibiotic use in modern medicine (25, 26). Agriculture and the distribution of agricultural products also enable increased rates of transmission of ARGs between soil and human microbiomes (27). Taken together, these factors suggest that agricultural soil ecosystems represent an important reservoir of ARGs relevant to humans (28, 29).

Indeed, the presence of clinically important ARGs, those that have been observed in clinical contexts, in pathogenic bacteria is higher in agricultural soil of all kinds compared to non-agricultural sites (30, 31). While a major driver of this high rate of clinically important ARGs is likely due to how agricultural soils are frequently exposed to antibiotics and enteric bacteria resistant to antibiotics (6, 19), other drivers may play key roles as well. Beyond exposure to antibiotic pollution, microbial communities in heavily farmed agricultural soils are exposed to extensive selective pressures exerted by intensive farming practices such as monoculture and the use of pesticides, herbicides, and fungicides, which disrupt the stability of soil microbiomes (32–34). Heavy metal concentrations have also been identified as highly correlated with antibiotic resistance, although this is independent of any specific land use, such as agriculture (35).

In other words, beyond antibiotic pollution, industrial agricultural methods including pesticides, heavy metals, and other chemicals present in settings give resistant invasive, opportunistic, or pathogenic soil bacteria an opportunity to flourish. For these reasons, it is important to understand the relationship between microbial biodiversity and the diversity of antibiotic resistance genes in the context of agricultural soils and industrial farming practices (36).

Here, we test the hypothesis that ARGs of clinical importance will be less abundant in soil harboring more diverse microbial communities. Specifically, we test the prediction that antibiotic resistance genes of clinical importance will be negatively correlated with bacterial diversity or with fungal diversity or both within a given ecosystem. Using *16S rRNA* and *ITS* amplicon sequencing to measure bacterial and fungal diversity respectively, in combination with whole-genome sequencing (WGS) to characterize the abundance of ARGs, we investigate the distribution of ARGs in relation to bacterial, fungal, and functional diversity along a single transect located in a hay field left untouched for 5 years on an otherwise active farm (Fig. S1). Importantly, while the whole transect has been left untouched for 5 years, the transect describes a gradient of increasing distance from an agriculturally active waterway and farm access road.

We find that bacterial diversity significantly increases along the agricultural transect, while fungal diversity has no significant change. Furthermore, functional diversity of identified genes is also correlated with increasing bacterial diversity but anti-correlated with fungal diversity, suggesting that the major driver of the diversity of function is the diversity of the bacterial population. Finally, considering clinically relevant and environmental ARGs, we find that they are not significantly correlated with each other or with the majority of functional genes. And while the diversity of environmental ARGs, like functional diversity, is significantly correlated with bacterial diversity and significantly

anti-correlated with fungal diversity, the diversity of clinical ARGs is only significantly anti-correlated with fungal diversity. This suggests that fungal diversity may be an important environmental factor in ARG diversity in general and in clinically relevant ARGs specifically.

## MATERIALS AND METHODS

### Soil collection

Samples were collected from four sites along a transect perpendicular to Esopus Creek, NY, as described in de Santana et al. (37) from Field 8 of Hudson Valley Farm Hub (Fig. S1), a hay field seeded in 2017 with a custom seed mix composed of Perennial Rye, Annual Rye, Timothy, Orchard Grass, and Huia Clover (a White Clover). Prior to this, the field was planted with oats and clover in 2016; supported vegetables in 2015, and earlier records suggest a history of sweet corn cultivation pre-dating its integration into the Farm Hub system. Management practices for the hay field have included annual fall mowing, with the exception of 2023, when mowing was deferred to address habitat considerations for avian species.

The four sites, chosen along a transect in increasing distance from a waterway, were labeled as follows: F8-W, a forested strip between the riverbank and a dirt road used by farming equipment; F8-0 (3.05 m), located in the hay field immediately next to the dirt road and about 15 feet from F8-W's forest edge. F8-300, located 300 feet (91.44 m) west of the forest edge and F8-600, located 600 feet (182.88 m) into the hay field. At each site, we collected three samples of 10 g topsoil (~5 cm deep) each, from sample sites located within 1 m of each other running parallel to the stream using sterile techniques. We sampled the transect three times over the spring and summer: 6 June, 19 June, and 3 July 2019. Samples were transported to the laboratory on ice in a dark cooler and frozen for at least 24 h before being processed. We geolocated one point along the transect at each distance to ensure that the exact locations were being sampled each time using GPS coordinates.

### DNA extractions and sequencing

We extracted and purified DNA from 1.8 g of soil for each sample using the Quick-DNA Fecal/Soil Microbe Miniprep kit (Zymo Research, Irvine, CA, USA). We then used this DNA for *16S rRNA* and internal transcribed spacer (*ITS*) amplicon sequencing as well as whole-genome sequencing (WGS). The V4 region of the *16S rRNA* gene was amplified using the 515F and 806R primers as per the Earth Microbiome Project (38). For ITS amplicon sequencing, we used the ITS1f and ITS2 primers (39). PCR products for *16S rRNA* and *ITS* amplicons were pooled separately and purified on a 2% agarose gel using the Qiagen gel extraction kit. As for whole-genome sequencing (WGS), libraries were prepared using the Nextera XT DNA Library Preparation kit (Illumina) and gel-purified. All purified libraries were quality checked using an Agilent 2100 Bioanalyzer and DNA High Sensitivity kit and pooled in an equimolar ratio. Purified pools were then stored at −20°C until sequencing. Amplicon libraries were sequenced at Wright Labs, Huntington, PA, using Illumina MiSeq v2 paired-end sequencing (2 × 250 bp reads) with 20% PhiX spike-in. WGS libraries were also sequenced at Wright Labs using an Illumina NextSeq 2000 (2 × 150 bp paired-end reads) with default parameters.

### Amplicon reads pre-processing and analysis

For *16S rRNA* and *ITS* amplicons, demultiplexed sequence reads were filtered and trimmed with Trimmomatic (40) (ILLUMINACLIP:TruSeq3-PE.fa:2:30:10 LEADING:3 TRAILING:3 SLIDINGWINDOW:4:15 MINLEN:100) with the requirement of a minimum average read quality score of 15 for inclusion. For each read, the sliding window cuts any read at the point where the median quality score over a 4-nt window is less than 15. To identify amplicon sequence variants or ASVs, that is unique identifiable sequences, we

used the QIIME2 pipeline (v2021.2) with default parameters except for DADA2 (16S rRNA, denoise-paired, –p-trim-left-f 0 –p-trim-left-r 0 –p-trunc-len-f 250 –p-trunc-len-r 250; ITS, denoised-single, –p-trunc-len 150). All ASVs were retained in the data set. Filtering was performed only on taxa and only for differential abundance analysis. Taxonomic assignment was performed using QIIME2's naive Bayes scikit-learn classifier (41) trained using the 16S rRNA gene sequences in the SILVA database (Silva SSU 138) (42) and UNITE's dynamic all taxa database v8.3 for ITS (43). Two ITS samples, F8-300 (2019-06-19) and F8-600 (2019-07-03), had too few reads and were removed from further analysis (Table S1).

Using ASVs as the unit of taxonomic identification, we estimated the total number of observed taxa (Table S2). For the purposes of alpha- and beta-diversity tests, we used QIIME2 rarefied 16S (max = 10K) and ITS (max = 350K) abundances. For all alpha-diversity comparisons between sites, significance was evaluated using Kruskal-Wallis (KW) statistics with adjusted $P$-values (Table S3). For all beta-diversity comparisons between sites, significance was evaluated using PERMANOVA statistics (Table S4).

We characterized the variance between time points and found no overall statistical effect of sampling date on bacterial diversity, measured either as the total number of observed bacterial ASVs ($H_{(2)}$ = 2.14; $P$ = 0.34) or Shannon's diversity index ($H_{(2)}$ = 0.63; $P$ = 0.73). Furthermore, we did not observe statistical differences between bacterial population structure at different time points using Bray-Curtis dissimilarity index (adonis: pseudo-$F_{(2,30)}$ = 1.0743; $R^2$ = 0.06; $P$ = 0.32). We also did not observe significant differences between dates in the total number of observed fungal ASVs ($H_{(2)}$ = 0.3; $P$ = 0.98) or Shannon's diversity index ($H_{(2)}$ = 2.13; $P$ = 0.34). Fungal community structure between sampling dates, based on Bray-Curtis dissimilarities, also did not change significantly (adonis: pseudo-$F_{(2,30)}$ = 0.7167; $R^2$ = 0.05; $P$ = 0.88). With this degree of similarity in mind, we treat these time points as biological replicates hereafter.

Furthermore, we used the Phyloseq package (44) for advanced analysis of sample subsets and specific taxonomy levels, and ggplot2 (45) for the visualization of data. We used the Vegan package (46) for the diversity analysis of prokaryotic and fungal communities as well as ARGs and functions. Vegan was also used for permutation analysis to identify differences between taxonomic and functional structures, as well as the resistome at each site. We used the Kruskal-Wallis test in R (version 4.2.1) (47) to check for statistical differences in the abundance of ARGs between collection sites and as an effect of the water influence. Correlation analysis was performed using in-built functions in the R environment using linear model ("lm") on the data of diversity.

## Whole-genome sequencing read processing and analysis

In order to access the taxonomic and functional profiles of the prokaryotic communities in the whole-genome sequencing, we used the MGnify 5.0 pipeline (48) from the European Nucleotide Archive (ENA) online platform (https://www.ebi.ac.uk/ena). Reads were assembled with metaSPAdes (v3.15.3) (49). Prediction of genes was performed using Prodigal (v2.6.3) (50) and FragGeneScan (1.20) (51), while functional annotation was performed with InterPro (v75.0) (52), a custom implementation of Gene Ontology (53), and Kegg Ortholog (v90.0) (54) using Kofam (v2019-04-06) (55). Individual proteins were then compared against UniRef90 (v2019_11) (56) using DIAMOND (v0.9.25.126) (57).

Furthermore, the WGS sequences were assembled using MEGAHIT (58) on the Galaxy platform (59), and the resulting contigs were then downloaded and locally compared with the CARD database (60) using the RGI tool in order to calculate the ARGs present within these contigs. In addition, we used a previously published study (61) to classify the health risk of all ARGs to assign high-risk, clinically relevant ARGs (e.g., Q1) or low-risk ARGs (Q2+). The latter is indicative of resistance genes common in the environment (here, referred to as "environmental ARGs") and unlikely to be linked to clinical issues, while the former indicates resistance genes currently linked with clinical implications for health (here, referred to as "clinical ARGs").

To visualize the correlation between ARGs and predicted gene function, we first subset both groups to only include instances that are present in a minimum of four samples (build_linregression.py). We next ran a linear regression on all pairwise combinations of normalized abundances, generated by RGI for the ARG or InterPro for gene function, the resultant *P*-values were then corrected using Benjamini-Hochberg (BH). The heatmap shows those pairwise combinations with BH adj. *P*-values ≤ 0.05.

Similarly, we accounted for antifungal resistance genes using RESfungi (62) to scan for the predicted protein coding sequences identified in the WGS data. This identified 320 likely anti-fungal resistance genes (E-value < 0.05) across seven categories. We then used linear regression (as above) to determine if any of these categories had significant changes in abundance relative to distance. Only the category of antifungal resistance genes, Azoles, was found to significantly vary in respect to distance, showing a slight but significant (*P*-value < 0.05) decrease over distance (Fig. S2) suggesting that antifungal resistance is not a likely confounding factor in metrics involving distances.

## Quantifying the abundance of specific antibiotic resistance genes

The DNA extracts were analyzed by quantitative real-time PCR (qPCR) on a CFX96 Touch Real-Time PCR Detection System (Bio-Rad, Hercules, CA) according to methods described in full elsewhere (63). In brief, each individual reaction was performed with SsoAdvanced Universal SYBR Green Supermix (Bio-Rad). In the present study, the gene targets were *16S rRNA*, *aadA*, *ampC*, *bla*$_{CTX-M}$, *erm*(B), *intI1*, strA, *sul1*, *tet*(M), and *vanA*. The qPCR runs included a standard curve covering seven orders of magnitude, and each sample was analyzed in duplicate. Standards were created using gBlocks Gene Fragments (Integrated DNA Technologies, Coralville, Iowa, USA). Primers and annealing temperatures for *bla*$_{CTX-M}$, *erm(B), intI1, sul1,* and *tet(M)* are described in Dungan et al. (63); conditions for genes *aadA*, *ampC*, *strA,* and *vanA* are described in Table S5. After estimating the copy number for each gene using the standard curves, we calculated the relative abundance of each gene by dividing their copy number by the copy number of *16S rRNA*, the latter being a bacteria universal gene used as a proxy for total bacteria count. For each gene present at all sampling sites, we performed statistical analysis in *R* by comparing models resulting from analysis of variance and linear regression, using the gene relative abundance as the dependent variable and sites as the independent variable.

## Measuring heavy metal concentrations in soil samples

To evaluate the potential impact of pollution on Field 8, we measured the concentration (mg/kg) of eight heavy metals: arsenic (As), barium (Ba), cadmium (Cd), chromium (Cr), copper (Cu), nickel (Ni), lead (Pb), and zinc (Zn) using three replicates at all four sites and for two depths, i.e., 5 and 15 cm. First, ~10 g of each sample was sent to the Cornell Soil Health Laboratories (Ithaca, NY) for standardized analyzes (64). Then, we used another ~5 g from the same collected soil to measure the concentrations of mercury (Hg) using an in-house procedure. In brief, dried soil samples were ground with a mortar and pestle, then sieved using a USS #10 sieve. Approximately 1.0 g of soil (dry weight) was added to a 250 mL round bottom flask and refluxed for 15 min with 2.5 mL conc. HNO$_3$ and 10 mL conc. HCl. After filtration through Whatman No. 41 filter paper, the filtrates were collected in 100-mL volumetric flasks. The filter was washed with 5 mL hot (~95°C) conc. HCl and 20 mL hot reagent water, with the washings added to the same flask. Residues and filters were returned to the flask, refluxed with 5 mL conc. HCl and reheated at 95°C until the filter dissolved. The solution was filtered again, cooled, and diluted to volume. The samples were centrifuged at 4,600 rpm for 5 mins, and the supernatants were transferred to scintillation vials. Heavy metal concentrations (mg/kg) were determined using the Agilent 5100 ICP-OES system (Agilent Technologies, Santa Clara, CA) after creating a standard curve. We then compared the concentrations of metals between the sites using analysis of variance, using the Tukey test for detecting differences between sites, when including the F8-W site. After removal of the F8-W site as an outlier, linear regression was used for testing the gradient of concentrations in the hay

sites (Fig. S4). We used a nested ANOVA to confirm that there was no difference in metal concentrations between samples taken at the surface, i.e., 5 cm, and soil samples at 15 cm.

## RESULTS

### Preliminary analysis of an unused hay field

To investigate the distribution of antibiotic resistance genes across the transect, we first sought to characterize the distribution of nine common antibiotic resistance genes relative to the abundance of the *16s rRNA* gene using qPCR. From the three genes that were present in all investigated sites, we found that both *aadA* ($F_{(1,7)} = 20.16$; $P = 0.003$; Fig. S3A) and *bla*$_{CTX}$ ($F_{(1,7)} = 8.39$; $P = 0.02$; Fig. S3B) decreased in relative abundance with distance, while *vanA* did not show any correlation with distance ($F_{(1,7)} = 0.09$; $P = 0.77$; Fig. S3C). This trend, combined with the fact that overall microbial diversity is considered a sign of soil health within an ecosystem, suggests that there might exist a gradient of environmental pollution within the transect, with environmental pollution predicted to be higher near the field's edge located close to an access road and the creek.

To address this, we looked for the presence of nine heavy metals often associated with pollution. Heavy metals do not degrade over time, and unlike other contaminants, they accumulate in soil and in living organisms. After analyzing the soil content for heavy metal concentrations, we observed that, indeed, three out of nine metals were slightly but significantly higher near the edge of the field and decreased linearly with distance from the edge: arsenic ($F_{(1,10)} = 28.70$; adj-$P < 0.001$; Fig. S3A), copper ($F_{(1,10)} = 87.27$; adj-$P < 0.0001$; Fig. S3C), and lead ($F_{(1,10)} = 26.23$; adj-$P < 0.001$; Fig. S3D). In all other cases, the heavy metals did not show any significant change with distance. Interestingly, we also found that the metal concentrations at the site near the water, i.e., F8-W, were similar to that of the site near the field's edge (Tukey test; adj-$P$s < 0.5), except for copper, which was lower near the water and lead, which was higher (Tukey test; adj-$P < 0.5$). Taken together, these results suggest that there indeed exists an environmental gradient within the transect, likely correlated with traces of environmental pollution in the soil.

### Microbial diversity and community structure differ among sites along the transect

We then investigated the microbial diversity and community structures among the transect sites. We found that the average bacterial diversity significantly differed between sites (Pielou's evenness, KW, $P = 0.014$; Fig. 1A). The highest average Pielou's diversity index, accounting for taxa evenness in addition to the number of taxa, was found at site F8-600, located 600 ft into the hay field, indicating that the latter showed a more even distribution among bacterial ASVs than any other sites (adj-$P$s < 0.05).

We also found that bacteria community structures (Fig. 1B) differed significantly between sites (Bray-Curtis, PERMANOVA; $P = 0.001$; Supplemental ST4) as well as each site in a pairwise comparison (pairwise, PERMANOVA: adj-$P$s < 0.01), suggesting that these bacterial communities are distinct. Notably, the F8-W site had a median 20.5% similarity (similarity = 1 − Bray-Curtis index) with the other three sites (Supplemental ST4) compared with a median of 36.3% within the other three sites, potentially because of the ecosystem differences between the forested F8-W site and the three hay field sites. In fact, when looking at community structure, communities found at each site along the transect were more closely related to each other, with an average 42.1% similarity, than to any other communities at other sites, with an average 27.8% similarity between the sites, suggesting that microbial communities might be impacted by an environmental gradient. Following taxonomic assignment, we found that the phyla Proteobacteria (median 33%) and Actinobacteria (median 20%) were the most abundant groups in all the collection sites.

Unlike what we found with bacterial communities, we did not find significant changes in Pielou's evenness index for fungal ASVs (KW, $P = 0.433$; Fig. 1C). When comparing

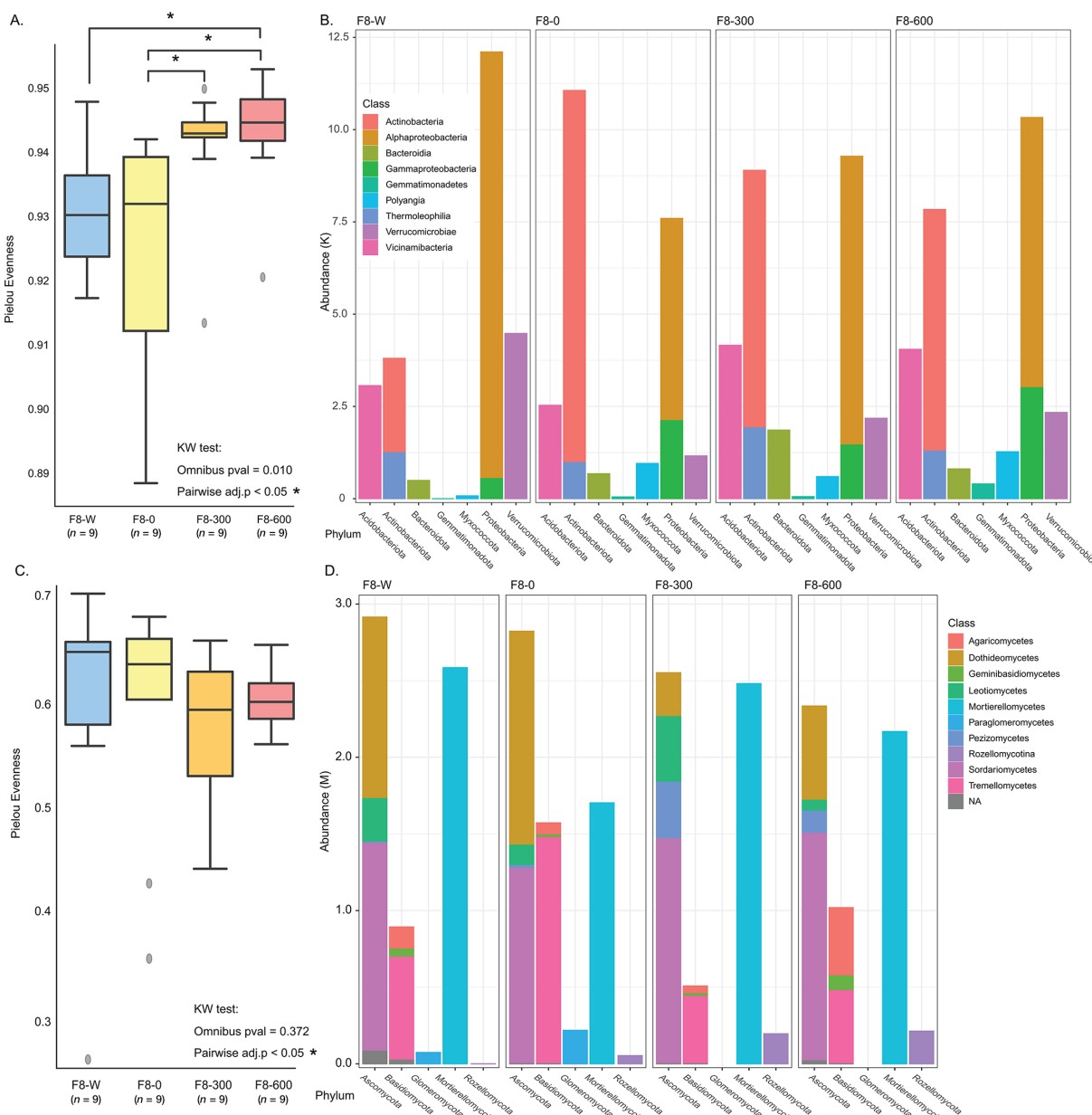

**FIG 1** Description of microbial communities along the F8 agricultural transect. 16S rRNA diversity in each sample as measured by (A) Peilou's evenness, significantly different pairwise sites (KW, BH adj.*P* < 0.05) are indicated by an asterisk. (B) Taxonomic box plot of the nine highest abundance 16S classes, separated by phylum. Its diversity in each sample as measured by (C) Peilou's evenness (no significant difference) (D) Taxonomic box plot of the 10 highest abundance fungal classes separated by phylum.

fungal community structures, however, we found significant differences between the four sites (Bray-Curtis:, PERMANOVA, *P* = 0.001; Supplemental ST4). Using the Bray-Curtis dissimilarity index, we found that sites differed significantly from each other by an average of 62%, mainly due to significant differences between the sites F8-W and F8-300, suggesting again that F8-W is more distinct. We found that phyla Ascomycota (median 41%) and Mortierellomycota (median 26%) were the most abundant groups in all sites (Fig. 1D).

As the preliminary analysis showed that the site F8-W was the most distinct for both the fungal and bacterial communities, we consider that the forested state and the close proximity to the water and the potential contamination present therein of herbicide, pesticide, and fertilizer runoff, in addition, F8-W is a forested site, which makes it a

significant outlier, and therefore, this site is held out from the remainder of this analysis. In addition to changes in diversity indexes in relation to distance, we also sought to identify genera whose abundances correlated strongly with distance (Fig. S5). While we did identify numerous genera with statistically significant changes in abundance in a linear relationship to distance, the vast majority of these had very small effect sizes (e.g., change in relative abundance < 1% global abundance).

## Functional diversity differs among sites along the transect

We also investigated total genetic diversity patterns using genes predicted from whole-genome sequencing (WGS) data. Overall, we found 6,924 functions belonging to 207 metabolic pathways. We found the most common functions to be associated with transposase activity, along with common metabolic processes, such as carbohydrate metabolic process and response to stress. We also found a number of metabolic pathways that are strongly associated with soil microbial communities, such as regulation of nitrogen use, which has a great impact on soil fertility and ecosystem functioning, as the plants are incapable of nitrogen fixing.

An analysis of functional diversity showed that the number of observed genes differed between the three sites (Fig. 2A). The genetic composition also showed significant differences between the sites, considering all sites (Bray-Curtis, PERMANOVA test: $P = 0.035$). When looking at functional diversity as a linear function of distance, we found that the number of different observed functions increased with distance ($F_{(1,7)} = 7.62$; $R^2 = 0.45$; $P = 0.02$; Fig. 2A). Similarly, we found that functional diversity, measured as the Shannon's diversity index, increased significantly with distance away from the field's edge ($F_{(1,7)} = 10.01$; $R^2 = 0.53$; $P = 0.02$; Fig. 2B). We also considered if functional diversity was correlated with either bacterial or fungal diversity, and while we did not find any statistically significant correlations between bacterial and functional diversity, we did find a significant negative correlation between functional diversity and fungal diversity (Fig. S6), suggesting that fungal diversity may play a role in functional diversity.

Finally, we also tested for linear trends in all functions individually. Overall, 16 functions were significantly correlated with distance (Wald Test, BH adj-$P \leq 0.05$). Broadly, we found the majority (69%) of these functions to be associated with genes known to provide antibiotic resistance or mobile genetic elements (MGEs, 13%) (Table S6). Three functions (18%) had no established association with either antibiotic resistance or MGEs, all of which were negatively correlated with distance (Fig. 2D). Notably, five antibiotic-associated functions are associated with riboflavin biosynthesis or flavin mononucleotide (FMN) or riboflavin-induced sensitization (65–67), suggesting that some antibiotic present in the environment may be positively correlated with distance.

## Distribution of antibiotic resistance genes among sites along the transect

We next sought to investigate the distribution of antibiotic resistance genes across the transect. To do so, we further investigated the whole-genome sequencing data to look specifically for predicted antibiotic resistance genes (ARGs), with specific interest in high-risk, clinically relevant ARGs (clinical ARGs) that have been previously identified (61) as compared with lower risk, non-clinical ARGs (environmental ARGs). Overall, we found 231 classes of ARGs, including 21 genes identified as high-clinical risk belonging to five classes. We found that the number of environmental ARG only changed marginally between sites ($H_{(2)} = 5.4$; $P = 0.06$; Fig. 3A, blue), and the same was observed when considering ARG diversity estimated with Shannon index ($H_{(2)} = 5.6$; $P = 0.06$; Fig. 3B, blue). Similarly, we found that the number of clinical ARGs did not change between sites ($H_{(2)} = 1.28$; $P = 0.53$; Fig. 3A, red) nor showed different levels of diversity ($H_{(df)} = 1.15$; $P = 0.56$; Fig. 3B, red).

We then tested whether individual ARGs correlated with distance from the field's edge by fitting a linear trend in the three hay field sites. We found 16 ARGs, detected in the majority of samples, to be significantly correlated with the transect distances (Wald Test, adj-$P \leq 0.05$; Fig. 3C). All of these significant ARGs that correlated with the distances

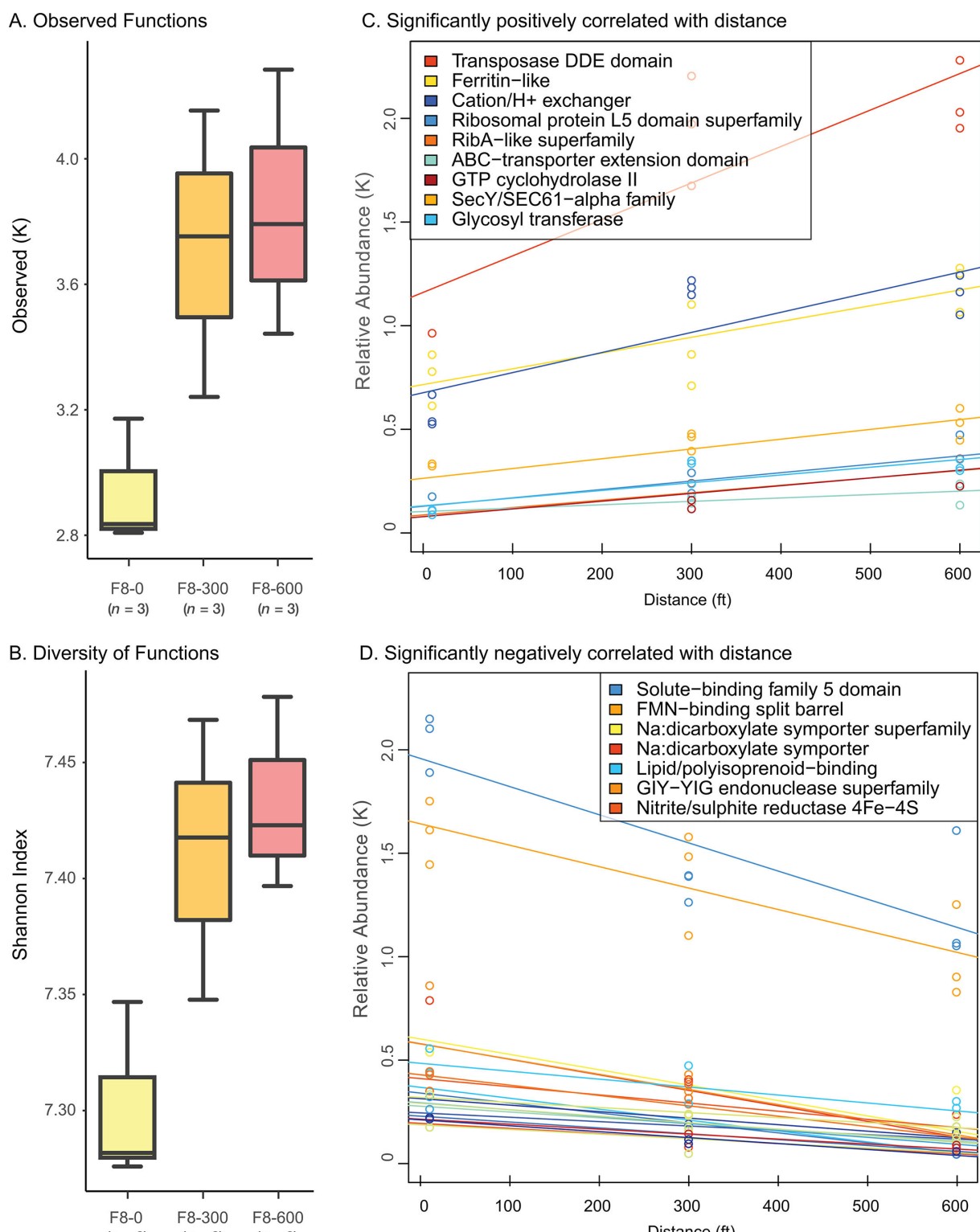

**FIG 2** Description of genetic diversity along the F8 agricultural transect. Genetic diversity in each sample, i.e., individual black circles, is represented as (A) the number of observed predicted genes, and (B) Shannon diversity index in the soil samples per distance in the transect. Relationships between predicted individual functions and distance along the transect in the hay field were analyzed using linear regression and found to be (C) positive ($P < 0.0001$) and (D) negative ($Ps < 0.0001$), where individual circles represent data points for each gene at each site.

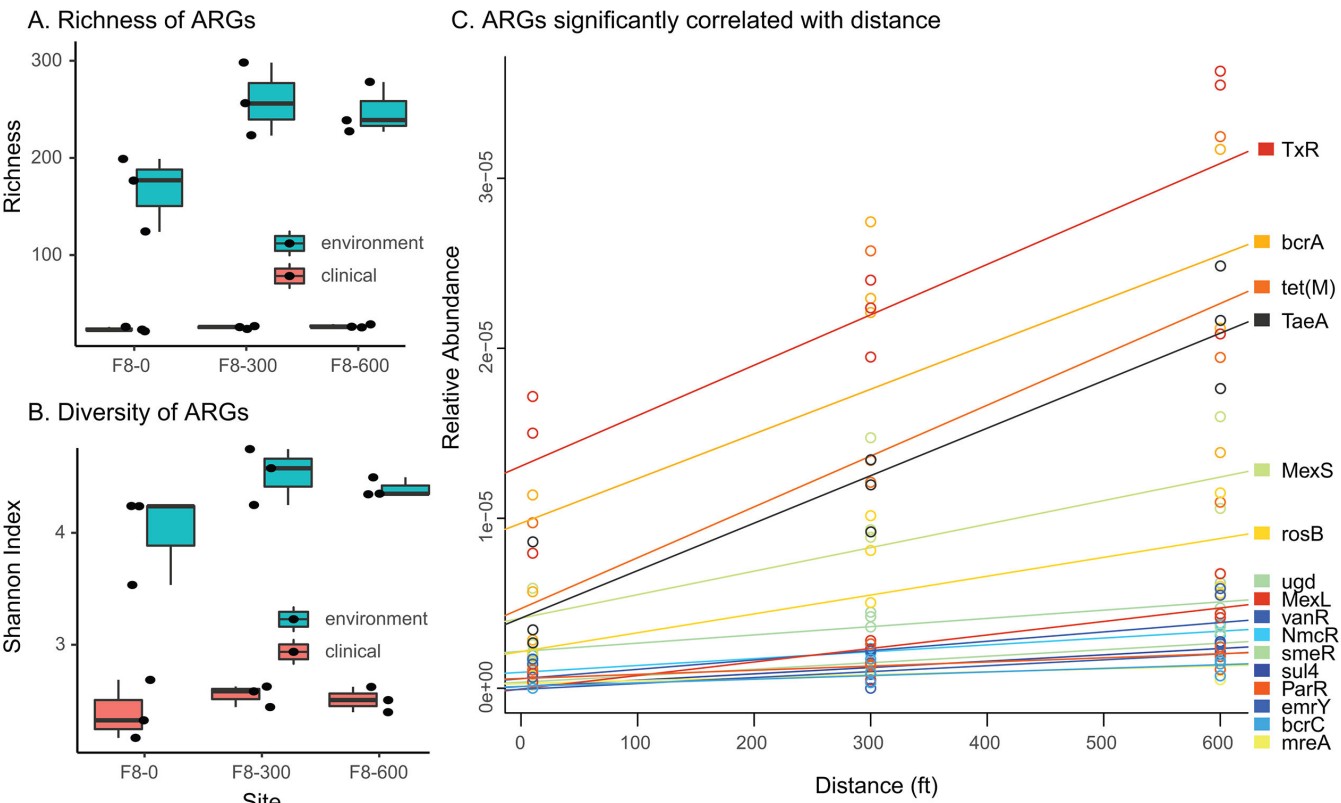

**FIG 3** Description of antibiotic resistance diversity along the F8 agricultural transect. Total antibiotic resistance genes (blue) and clinical ARGs (red) diversity in each sample, i.e., individual black circles is represented as (A) the number of observed predicted ARGs genes, and (B) Shannon diversity index in the soil samples per distance in the transect. Blue shade shows values in the wooded samples nearer to the water. Data for each sampling date are presented separately but were analyzed as biological replicates. Relationships between predicted individual functions and distance along the transect in the hay field were analyzed using linear regression and found to be (C) positive ($P < 0.0001$), where individual circles represent data points for each ARG at each sample site.

in the transect increased relative to the distance from the road. Only three of these genes, *tetM*, *ugd*, and *emrY,* are clinical ARGs, 1.2-fold lower number of observations than what would be expected by chance but not significantly so (FET, $P = 0.83$).

## Investigating relationships between antibiotic resistance and microbial diversity

One of the main objectives of this study was to investigate the distribution of ARGs across the transect in relation to bacterial and fungal diversity, especially when it comes to clinically relevant resistance genes. Based on previous research, we predicted that more diverse soil communities would correlate with a reduction in clinically relevant ARGs.

We first investigated whether bacterial communities' diversity could explain, at least in part, the distribution of antibiotic resistance in our samples. We found that neither the environmental ARG abundance ($F_{(1,6)} = 2.00$; $R^2 = 0.44$; $P = 0.21$; Fig. 4A) nor the clinical ARG abundance correlated with the total number of bacterial ASVs ($F_{(1,6)} = 0.24$; $R^2 = 0.14$; $P = 0.64$; Fig. 4A). Similarly, we found that neither the environmental ARGs ($F_{(1,6)} = 0.79$; $P = 0.79$; Fig. 4B) nor the clinical ARGs correlated with the total number of fungal ASVs ($F_{(1,6)} = 3.53$; $P = 0.11$; Fig. 4B).

However, unlike abundances, we did find that environmental ARG diversity positively correlated with bacterial diversity ($F_{(1,4)} = 14.46$; $P = 0.01$; Fig. 4D) as well as with fungal diversity ($F_{(1,5)} = 22.66$; $P = 0.005$; Fig. 4E). However, we did not find the same pattern when considering clinical ARG diversity relative to bacterial diversity ($F_{(1,4)} = 0.12$; $P = 0.12$; Fig. 5D) and found that clinical ARG diversity slightly, but significantly, decreased as

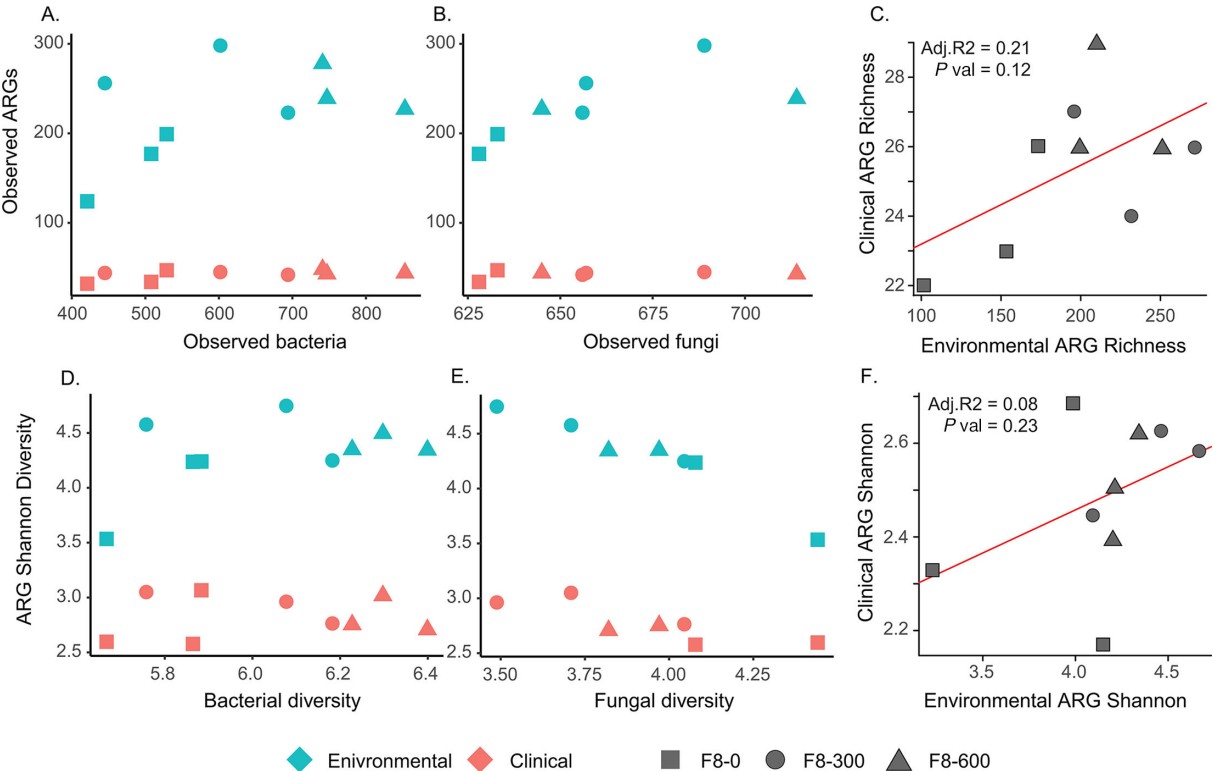

**FIG 4** Correlations between antibiotic resistance gene diversity and microbial diversity. Total number of predicted environmental ARGs (blue) and clinical ARGs (red) is compared with (A) the observed number of bacterial ASVs, (B) the observed number of fungal ASVs, and (C) the correlation between clinical and environmental ARG richness. Also, the Shannon diversity of predicted environmental ARGs (blue) and clinical ARGs (red) is compared with the diversity of (D) the bacterial ASVs, (E) fungal ASVs, and (F) the correlation between clinical and environmental ARG diversity indexes.

fungal diversity increased ($F_{(1,5)}$ = 13.08; $P$ = 0.01; Fig. 4E). In other words, these results suggest that while there might be a positive relation between environmental ARG diversity and both bacterial and fungal diversity, this relation does not exist for clinical ARGs.

Finally, we wanted to directly test the possibility that environmental ARGs had a direct correlation with clinical ARGs. However, we did find neither the Observed number of ARGs ($P$ = 0.12, Fig. 4C) nor the Shannon diversity indexes ($P$ = 0.23, Fig. 4F) was significantly correlated.

Given that some classes of antibiotics are commonly produced and secreted into the environment by fungi, we sought to find instances of ARGs that are significantly correlated with fungal genera. Using a linear regression of the relative abundances of both ARG family and fungal genera, we found genes encoding resistance to beta-lactam antibiotics, a class of antibiotics that include penicillin and cephalosporin commonly produced by fungi. However, while we did not find any taxa known to produce beta-lactam, we found other genera, such as *Fusarium* (68), that are known to produce beta-lactamases (Fig. S7). This suggests that the observed correlation is not caused by the production of beta-lactam by fungi but by a resistance to beta-lactam shared between these ARGs and some fungi.

## Correlations between ARGs and functional diversity

Since some genes may function in a way that encourages ARGs to arise and be selected for, we next sought to determine if any strong correlation may be observed between individual ARGs and gene functions. To do so, we performed a pairwise analysis of the correlation between relative gene abundances and relative ARG abundances. While

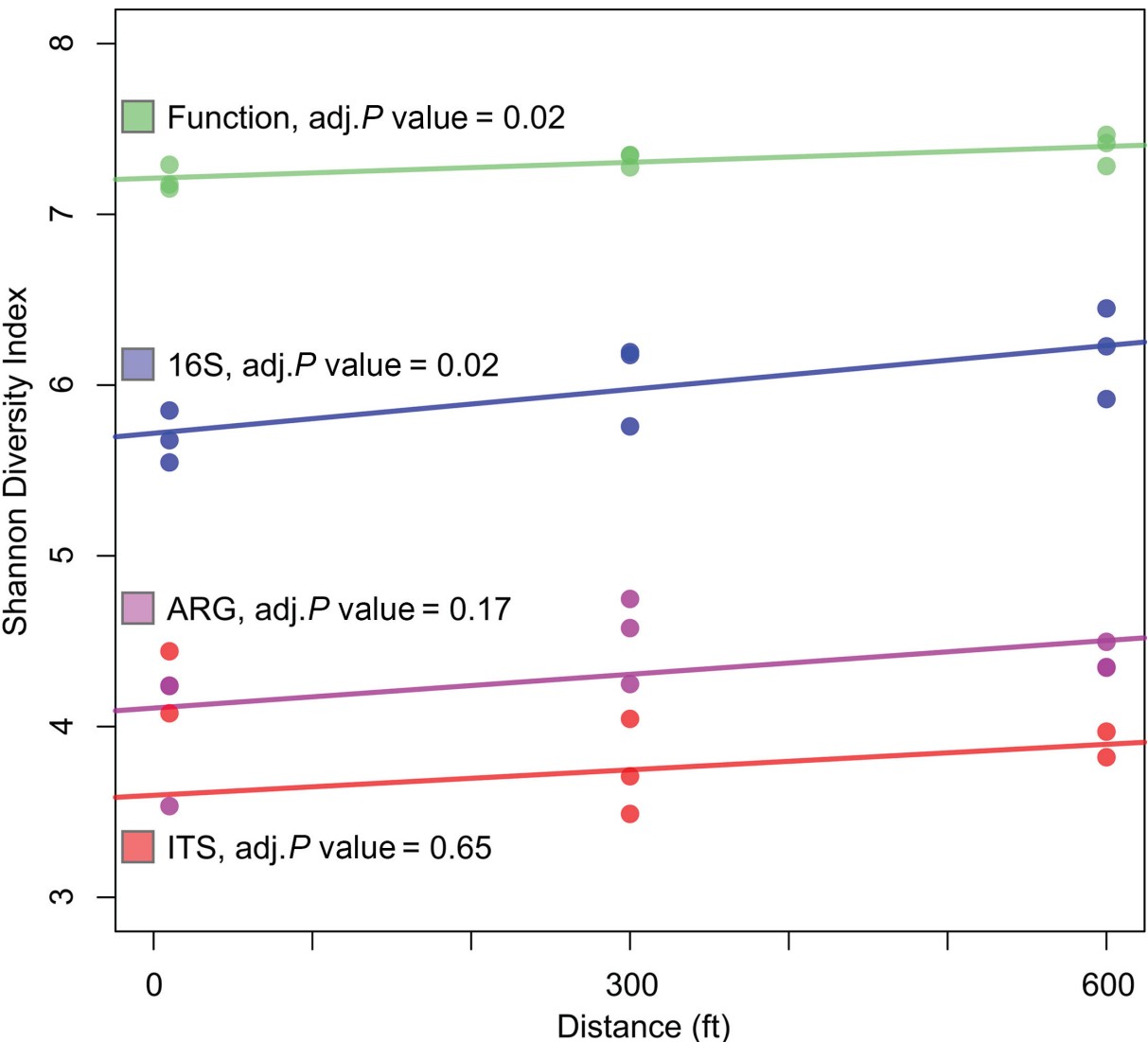

**FIG 5** Linear trends of measured diversity indices over distance along the F8 transect. Linear trends, estimated using linear regression, are represented for diversity in bacterial ASVs (blue), fungal ASVs (red), predicted function genes (green), and antibiotic resistance genes (purple). Each point represents data from an individual sample.

some structure is present (Fig. S8), the vast majority of interactions (~98.9%) were not statistically significant ($Ps > 0.05$). Yet, we observed a few (~1.1%) significant correlations (Benjamin-Hochberg adj-$Ps \leq 0.05$), including four clinically relevant ARGs (Fig. S7). The correlations included strongly positive relationships, such as the cytochrome p450 involved in the cell stress response system with *mexY*, an important ARG found in pathogenic bacteria. We also found strong positive correlations such as one between tetracycline resistance genes and the DNA-binding HTH domain TetR-type, a domain of proteins known for its role in the resistance to this antibiotic class. Taken together, our results suggest that the significant correlation between ARGs and gene diversity is not a spurious result of correlated abundance.

## Investigating the effect of distance within the transect on microbial communities

When considering the general effect of the transect covering 600 feet (~183 m) of unused hay field, we observed a general trend suggesting that most indicators changed

in relation to distance from the field's edge. To test this hypothesis, we further investigated these trends using linear regression analyses. We found a significant positive linear relationship between distance from the creek and both functional and bacterial diversity (Fig. 5). On the other hand, although slightly positive, we did not find ARG or fungal diversity to have a significant linear relationship with distance.

## DISCUSSION

Bacterial infections resistant to antibiotics are an important public health issue and are predicted to become one of the main causes of mortality globally by 2050 (69). Because antibiotic resistance genes (ARGs) are naturally present in microbial communities from environmental reservoirs (17, 20, 25, 26, 70), the World Health Organization as well as the Centers for Disease Control and Prevention are advocating for a One Health approach that seeks to consider the important links between human health, animal health, and environmental health to manage the growing antibiotic resistance crisis (71). Thus, understanding the factors that shape the distribution of antibiotic resistance genes in the environment can provide key information to mitigate the emergence and spread of clinically relevant antibiotic resistance genes (13).

Here, we investigated how the diversity of antibiotic resistance genes related to bacterial and fungal diversity as well as total functional diversity in an old hay field along an agricultural transect located at the Hudson Valley Farm Hub, an active farm located in New York State. The aim of this study was to understand the role of bacterial and fungal diversity in shaping the presence of clinical and environmental antibiotic resistance genes.

When first looking at the total number of ARGs found in the environment, we found a positive correlation with bacterial, fungal, and functional diversity. Notably, we found that bacterial, functional, and environmental ARG diversity increased with increased distance from the agricultural water course and farm access road. While this trend is consistent with increasing soil health (36), it is not necessarily the only explanation (72, 73). Furthermore, it is important to note that comparing soil microbial diversity across different ecosystems presents challenges due to varying environmental conditions, which complicate direct comparisons of microbial diversity and its implications for soil health across ecosystems (74). This is doubly important given the small scale of this current study. Additionally, increased environmental ARG diversity may be a consequence of increased functional diversity as ARGs are both functions in themselves (75) and have often additional functions separate from antibiotic resistance, such as molecule transport via the cell membrane (76) or as signaling molecules at low concentrations (77). To account for this, we sought to determine the frequency at which ARG abundance significantly covaried with functional abundance, and we found that the vast majority (>98%) of functions were not significantly correlated with ARGs, suggesting that the observed trends of these two categories can be considered independent. This supports the independence of ARG analysis as separate from broader functional analysis and important considerations given the increasing availability of pipelines to document the presence of antibiotic resistance in metagenomics data (78–80).

However, when focusing on ARGs of clinical importance, we found that they did not increase with bacterial diversity. This result is interesting given that bacteria are the predominant carrier of antibiotic resistance genes. At least two mechanisms could explain such a finding. For one, many clinically relevant ARGs spread into pathogens after being mobilized on mobile genetic elements, such as plasmids, phages, and integrons, thus spreading horizontally under selective pressures imposed by higher antibiotic concentrations (81, 82), such as the case with some agricultural soils (83). Second, many antibiotic resistance genes also find their way into clinical settings by being acquired by opportunistic pathogens that tend to show higher adaptive abilities like exogenous DNA acquisition (84–86). While the two mechanisms are not mutually exclusive, both make it possible that changes in clinically relevant ARG distribution in response to local selective pressures could be significantly different than that of their bacterial hosts.

In addition, we also found that ARG diversity, both environmental and clinically relevant, was negatively correlated with fungal diversity (Fig. 5E). This is in contrast to some models of bacterial-fungal interactions that predict a positive correlation, given the widespread capacity of fungi to produce natural antibiotics (Fleming, 1929)(87, 88) and the expectation that bacteria exposed to a diversity of antibiotic-producing fungi would benefit from having a higher number of antibiotic-resistance genes (89, 90), and that fungi are known to actively help the horizontal transfer of ARGs between bacteria (89,91). However, this observation is consistent with other research that finds that fungi can change resistome structure (92), and that bacteria and fungi can present widespread antagonistic behavior in soil depending on the local environment (93); such that in sites with higher diversity of fungi, competition may be selecting bacteria with specific resistance genes rather than a generally diverse resistome (94). However, given the scale of this study and the limitations of *ITS* amplicon sequencing to document fungal taxonomy, future work, potentially combining a culture-based approach and genomics, will likely be needed to resolve this difference.

Importantly, while we observe that bacterial alpha-diversity changes with respect to distance, we do not observe a similar change in fungal diversity. Because we did not measure any environmental variable that could explain this observation, we suggest that this pattern could emerge from the biological and physiological properties of these two distinct kingdoms. Specifically, the life history traits that typify a bacterium may be substantially smaller and shorter in scale than those of a fungus. For example, dispersal distances for these two categories of microbes have been suggested to be an important driver in distance-decay relationships (DDR) (95, 96), which is a correlation between beta-diversity and spatial distance (97). Indeed, recent research on soil substrate in semi-arid farmland found substantial differences between bacterial DDR, which increased significantly with distance, and fungal DDR, which did not (98), consistent with what we observe here.

While our results suggest that promoting diverse fungal communities could be used to reduce the distribution of ARGs in the environment, our current study does not allow us to identify under what conditions this relationship holds. Fungal communities are known to be influenced by a complex interplay of factors such as soil properties, environmental conditions, and biotic interactions (89). For example, fungal communities can be more sensitive than bacterial communities to environmental variables like precipitation or nutrient availability (99). Furthermore, many fungi can have symbiotic relationships with plants, meaning that their distribution will be codependent on the latter (100). Given that plant diversity changes over different sections of the hay field, it is likely that fungal diversity was affected by other environmental factors at these sites. We plan to investigate the interaction between plant diversity, fungal diversity, and antibiotic-resistance genes in future work.

It is also possible that we did not find correlations between specific fungal taxa and ARGs due to the fact that we used a single-gene amplicon library strategy to estimate diversity. While this technique for studying microbiomes has been useful in a diverse array of studies, it is known that amplicon library sequencing underestimates genetic variability at the sub-genus level (101, 102). The latter may be especially relevant with antibiotic resistance genes, which are often exchanged horizontally within and among species (103). The fact that ARG diversity correlated with functional diversity, which used WGS, indicates that novel approaches, such as long-read sequencing, to estimate bacteria diversity might be better suited for such work in the future.

Finally, while the transect was originally designed to capture different ecological niches in relation to the field edges, our results suggest that there exists a disturbance gradient along the transect. Using heavy metal concentrations, we found that, on average, site F8-0 presented the highest concentration of metals, which then decreased with distance. Because some ARGs can confer cross-resistance to heavy metals (104), this could explain, at least in part, the fact that clinical ARGs were higher near the road. Indeed, *aadA*, *bla*$_{CTX-M}$, and *vanA* have all been shown to be co-selected with the

presence of heavy metals, especially copper in agricultural settings (105, 106). Therefore, it is possible that the presence of the Esopus Creek and the apparent gradient in heavy metal concentrations could provide a confounding explanation for the selection of antibiotic resistance correlating with microbial biodiversity. However, it is important to note that while the literature suggests heavy metals increase ARG abundance and diversity (107), this is not what we observed, where the highest concentrations of metal were close to the edge of the transect, while the highest ARG diversity was furthest from the edge. However, future work will be necessary to determine what mechanisms are more important.

Taken together, our results suggest that fungal diversity can influence the diversity of ARGs. This seems to occur in tandem with the bacterial diversity for environmental ARGs, but may be a dominant factor in the diversity of clinically relevant ARGs. Future work on ARG diversity and ARG environmental reservoirs could focus on factors that influence soil fungal diversity, such as farming practices.

## ACKNOWLEDGMENTS

This study is part of the Applied Farmscape Ecology Research Collaborative (AFERC). AFERC is co-coordinated by Hawthorne Valley Farmscape Ecology Program and The Hudson Valley Farm Hub and is funded by The Hudson Valley Farm Hub. This study was also funded by the Office of Undergraduate Research at Bard College and the Bard Summer Research Institute (BSRI).

The authors would like to thank Anne Bloomfield and Teresa Dorado for their assistance in managing the project at FarmHub, Christopher Benincasa, Hannah Herrick, and Tejaswee Neupane for their assistance in the field and DNA extraction and Chad McKinney for his work on qPCR.

## AUTHOR AFFILIATIONS

[1]Department of Exact Sciences, State University of Feira de Santana, Brasília, Bahia, Brazil
[2]Bard Center for Environmental Sciences and Humanities, Bard College, Annandale-on-Hudson, New York, USA
[3]Center for Genomics and Systems Biology, New York University, New York, New York, USA
[4]Hawthorne Valley Farmscape Ecology Program, Hawthorne Valley Association, Ghent, New York, USA
[5]Chemistry and Biochemistry Program, Bard College, Annandale-on-Hudson, New York, USA
[6]USDA-ARS, Northwest Irrigation & Soils Research Laboratory, Kimberly, Idaho, USA

## AUTHOR ORCIDs

Carolina Oliveira de Santana  http://orcid.org/0000-0002-9385-0199
Pieter Spealman  http://orcid.org/0000-0002-7105-284X
Conrad Vispo  http://orcid.org/0000-0003-3899-0620
David Gresham  http://orcid.org/0000-0002-4028-0364
Christopher N. LaFratta  http://orcid.org/0000-0003-4585-6278
Swapan S. Jain  http://orcid.org/0000-0002-7475-6754
Robert S. Dungan  http://orcid.org/0000-0002-7560-5560
Gabriel G. Perron  http://orcid.org/0000-0003-3526-5239

## AUTHOR CONTRIBUTIONS

Carolina Oliveira de Santana, Conceptualization, Data curation, Formal analysis, Investigation, Methodology, Visualization, Writing – original draft, Writing – review and editing | Pieter Spealman, Conceptualization, Data curation, Investigation, Methodology,

Visualization, Writing – original draft, Writing – review and editing | Conrad Vispo, Conceptualization, Writing – review and editing | David Gresham, Funding acquisition, Writing – review and editing | Sage Saccomanno, Formal analysis, Writing – review and editing | Christopher N. LaFratta, Formal analysis, Validation, Writing – review and editing | Swapan S. Jain, Formal analysis, Methodology, Writing – review and editing | Robert S. Dungan, Formal analysis, Writing – original draft, Writing – review and editing | Gabriel G. Perron, Conceptualization, Data curation, Formal analysis, Investigation, Methodology, Project administration, Visualization, Writing – original draft, Writing – review and editing

## DATA AVAILABILITY

All computational scripts used in this analysis of the data are publicly available as a versioned GitHub repository here: https://github.com/pspealman/transect/releases/tag/v0.1. All sequencing data have been deposited in both NCBI: PRJEB52998 and EBI: PRJEB54565.

## ADDITIONAL FILES

The following material is available online.

### Supplemental Material

**Supplemental Figures (Spectrum03232-25-s0001.docx).** Figures S1 to S8.
**Table S1 (Spectrum03232-25-s0002.txt).** Tab delimited file. Contains sequencing run quality control numbers for amplicon sequencing runs generated by denoising using DADA2.
**Table S2 (Spectrum03232-25-s0003.txt).** Tab-delimited file. Contains QIIME2 Feature IDs with taxonomic assignments and confidence scores.
**Table S3 (Spectrum03232-25-s0004.txt).** Tab-delimited file. Contains omnibus and pairwise alpha-diversity tests (Evenness, Faith-PD, Shannon) and beta-diversity tests (Jaccard, Bray-Curtis, Unweighted UniFrac, Weighted UniFrac) for both ITS (rarefaction = 350K) and 16S (rarefaction = 10K) runs.
**Table S4 (Spectrum03232-25-s0005.txt).** Tab-delimited file. Contains all pairwise Bray-Curtis dissimilarity (i.e. percentage difference) for all 16S and ITS samples.
**Table S5 (Spectrum03232-25-s0006.xlsx).** Condition for antibiotic resistance gene qPCR.
**Table S6 (Spectrum03232-25-s0007.csv).** Analysis of functions significantly correlated with distance, their Interpro ID, broad category, and association with antibiotic resistance or mobile genetic elements, if any.

### Open Peer Review

**PEER REVIEW HISTORY (review-history.pdf).** An accounting of the reviewer comments and feedback.

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
