## [Reviewer comments · Microbiology Spectrum]

Microbiology Spectrum

Investigating the distribution of antibiotic resistance genes in relation to bacterial, fungal, and functional diversity in a hay field.

Carolina de Santana, Pieter Spealman, Conrad Vispo, David Gresham, Sage Saccomanno, Chris LaFratta, Swapan Jain, Robert Dungan, and Gabriel Perron

Corresponding Author(s): Gabriel Perron, Bard College

Review Timeline:

Submission Date:	October 31, 2025
Editorial Decision:	December 19, 2025
Revision Received:	February 25, 2026
Accepted:	February 26, 2026

Editor: Ryan Blaustein

Reviewer(s): The reviewers have opted to remain anonymous.

Transaction Report:

DOI: <https://doi.org/10.1128/spectrum.03232-25>

Re: Spectrum03232-25 (Investigating the distribution of antibiotic resistance genes in relation to bacterial, fungal, and functional diversity in a hay field.)

Dear Dr. Gabriel G Perron:

Thank you for the privilege of reviewing your work. Below you will find my comments, instructions from the Spectrum editorial office, and the reviewer comments.

From the Editor: Thank you for addressing both of the initial reviewers' comments in your manuscript revision at the time of its transfer to mSpectrum. The revised manuscript was sent back out for expert review and only minor modifications are requested. Please see the attachment with comments from Reviewer 1.

Revision Guidelines

Sincerely,
Ryan Blaustein
Editor
Microbiology Spectrum

Reviewer #1 (Comments for the Author):

Overall, good work and interesting research!

This is an interesting topic. I liked the idea of investigating both clinical and environmental ARG diversity. The results and discussion are convincing after editing and adding more explanation and reasoning based on two reviewers' comments/suggestions. Overall, the authors did a good job incorporating reviewers' comments/suggestions into their research. The paper is scientifically well written and edited. There are a few grammatically points that I am going to point out below:

Introduction: (in the Marked-Up-Manuscript)

- Line 106: change it to: Indeed, the presence of clinically important antibiotic resistance ARGs which have been observed in clinical contexts, in pathogenic bacteria, is higher in agricultural soil of all kinds compared to non-agricultural sites.

Results: (in the Marked-Up-Manuscript)

- Line 355: change it to: located close to road and the creek.
- Line 402: change it to: in addition, F8-W is a forested site, which makes it a significant outlier, and therefore, this site is held out from the remainder of this analysis.
- Line 404: add “,”: In addition to changes in diversity indexes in relation to distance, we also sought to identify genera whose abundances correlated strongly with distance (Supplemental Figure S5).
- Line 406: add “,”: While we did identify numerous genera with statistically significant changes in abundance in a linear relationship to distance, the vast majority of these had very small effect sizes (eg. change in relative abundance < 1% global abundance).

We thank both the reviewers and the editor for their time and attention to our manuscript. We agree with the general consensus that the scope of the research would better position it for submission at ASM sister publication Spectrum. In support of this we have revised the manuscript to remove draft notes, typos, and to focus more on the experiment and direct observations, this includes several new supplemental components that address the reviewer's original concerns.

Reviewer #1

This study investigates the relationship between antibiotic resistance genes and microbial diversity in agricultural soils, specifically in a hay field, and tries to find correlations between clinically relevant antibiotic resistance genes and bacterial and/or fungal diversity.

The authors use a combination of 16S rRNA and ITS amplicon sequencing, metagenomics and quantitative PCR (qPCR) to investigate bacterial, fungal, and functional diversity along a transect in a hay field.

The findings provide valuable insights into the interplay between microbial diversity and both environmental and clinically relevant ARGs. However, there are several areas where the manuscript could be improved to enhance clarity.

These include:

- The introduction needs to be revisited and improved, as it looks like it is still in the draft stage. For example, "Define environmental reservoirs? Indeed..." appears in the introduction. Also, the authors could condense the discussion of antibiotic resistance in clinical settings and focus more on the environmental context which is more relevant to this study.
 - We have removed the draft not "Define environmental ..." and we apologize for its unintended inclusion.
 - We agree with the reviewers that the text was overly broad in trying to communicate both environmental observations and clinical interpretations simultaneously. We have narrowed the text to focus on the environmental context of the hay field.

- The methods could provide more detail. For example, including a figure with the sampling sites would make the results much easier to follow. Also, additional details on statistical analyses, including effect sizes and confidence intervals could be provided to improve reproducibility. Also, providing accession numbers for the data generated in this study would be welcomed.

- We agree with the reviewer and we have added a map of the sampling sites as Supplemental Figure 1.
 - We have clarified the statistics where possible but have retained the p -value in preference to confidence intervals as the former allows for a straightforward method of multiple hypothesis testing correction that can be applied uniformly across the various statistical tests we have performed.
 - We apologize that we had neglected to include accession codes in our original submission. We have corrected this oversight and have added them into the text.
- Regarding results, the negative correlation between fungal diversity and clinically relevant ARGs is very interesting and could be explored in more detail. Identifying specific fungal taxa associated with increased ARGs and check whether the ARGs are related to reported antimicrobials produced by those taxa would strengthen this argument.
 - We thank the reviewer for this important question. We performed the requested analysis and found that some ARGs that grant resistance to classes of antibiotics known to be produced by fungi, primarily beta-lactams, are positively correlated with relative abundances of some fungi. However, these fungi are not known to produce these classes; and some, in fact, are known to produce beta-lactamases that provide resistance to beta-lactams. We have included this as Supplemental Figure 7.

- The role of heavy metals as a confounding factor is mentioned but not thoroughly explored. The authors could discuss whether the observed trends in ARGs could be driven by co-selection with heavy metals rather than microbial diversity.

 - We agree with the reviewer that our text originally glossed over this important point and we have amended the text to include this additional information.
 - However, it is important to note that while the literature suggests heavy metals increase ARG abundance and diversity this is not what we observed, where the highest concentrations of metal were close to the edge of the transect while the highest ARG diversity was furthest from the edge.
- The authors identify 16 functions significantly correlated with distance. It would be interesting to discuss their ecological or functional relevance in more detail. A deeper exploration of these functions could provide insights into the mechanisms driving ARG distribution in this transect.

- We thank the reviewer for their prescient insight. We performed the analysis and found that nearly each of the 16 functions identified as significantly correlated with distance had some connection, often direct, with antibiotic resistance or horizontal gene transmission, now described as Supplementary Table 6. We have amended the text to include this observation:
- Lastly, we also tested for linear trends in all functions individually. Overall, 16 functions were significantly correlated with distance (Wald Test, BH adj.P \leq 0.05). Broadly, we found the majority (69%) of these functions to be associated with genes known to provide antibiotic resistance or mobile genetic elements (MGEs, 13%) (Supplemental Table S6). Three functions (18%) had no established association with either antibiotic resistance or MGEs, All of which were negatively correlated with distance (Figure 2D). Notably, 5 antibiotic associated functions are associated with riboflavin biosynthesis or flavin mononucleotide (FMN) or riboflavin induced sensitization suggesting that some antibiotic present in the environment may be positively correlated with distance.
- Finally, some citations (e.g., "[No Title]") appear incomplete and should be corrected.
 - We apologize for the formatting error and we have fixed the problematic citations
- Overall proofreading of the entire document is recommended to remove incomplete notes and comments as well as to improve flow and clarity.
 - We again apologize for the draft notes and comments that were present in the original text. We've removed them.

Reviewer #2

Santana et al in this manuscript investigated the how the diversity of antibiotic resistance genes (ARGs) related to bacterial and fungal diversity as well as total functional diversity in an old hay field. The authors found positive correlations of the total number of ARGs found in the environment with bacterial, fungal, and functional diversity. However, there was no association between clinically-relevant ARGs and functional diversity or bacterial diversity. This study also highlighted the potential impact pollution present in the transect by measuring the levels of heavy metals.

Comments:

- This study has a small number of samples, which were collected from a single hay field.
 - We understand the reviewer's concerns and we have revised the scope and the wording of the document to reflect the small size of this study.

- The number and scope of antibiotic resistance genes analyzed for abundance are relatively low. Moreover, if this study examines fungal diversity, it would be also interesting to investigate anti-fungal resistance genes present in the samples to look for associations with bacterial and fungal diversity.
 - We thank the reviewer for this very interesting question. We performed the additional analysis to identify potential anti-fungal resistance genes using the recently published Hmmer database RESfungi, we did find that a decrease in several broad classes of resistance genes negatively correlated with distance.
 - This have amended the text to include this observation and have added it as Supplemental Figure S2

- It could provide more insight if the authors can elaborate on the discussion about why the fungal community structures were not significantly different across the four sites.
 - We agree with the reviewer that this is an observation worth discussion. Notably, there is no specific environmental variable that we measured that would affect microbial diversity of bacteria but not fungi. However, given the very different life histories of these organisms, differences in spatial distances may arise from differences in reproduction, sporulation, and dispersal. The following text has been added to the discussion.
 - Potentially, however, we may consider that the scale of existence or life history that typifies a bacteria may be marginally smaller than that of a microbial fungi. Indeed, dispersal distances for these two categories of microbes has been suggested to be an important driver in distance-decay relationships. The distance-decay relationship (DDR) describes the correlation between beta-diversity and spatial distance. Indeed, recent research did find substantial differences between bacterial DDR, which increased significantly over distance and fungal DDR which did not change significantly over distance in semi-arid

farmland, which supports what we observe here.

- "This suggests that farming practices that favor fungal diversity could result in the reduction of antibiotic resistance genes spread in the environment" The interpretation seems too far-fetched based on the findings from this study.
 - We agree with the reviewer and we have removed this speculation from the document.
- Suggest adding line numbers to aid the review process.
 - We apologize for the oversight and we have added line numbers to the document.
- "Define environmental reservoirs?" can be deleted.
 - We have deleted this and other comments from the text and we apologize for their inclusion.
- "While it is possible that under certain conditions soil could show high levels of diversity due to the "Intermediate Disturbance Hypothesis" (Connell 1978; X. Zhang et al. 2018), healthy soils are generally characterized by a high diversity of bacterial and fungal microorganisms (Yoon et al. 2024)" Is it referring to under certain conditions soil could show LOWER levels of diversity?
 - We agree that the original text was not clear. We were trying to say that multiple models present predict conflicting outcomes. In re-writing this we realized that it was rather outside the scope of the paper and we've since removed section.

Re: Spectrum03232-25R1 (Investigating the distribution of antibiotic resistance genes in relation to bacterial, fungal, and functional diversity in a hay field.)

Dear Dr. Gabriel G Perron:

Your manuscript has been accepted, and I am forwarding it to the ASM production staff for publication. Your paper will first be checked to make sure all elements meet the technical requirements. ASM staff will contact you if anything needs to be revised before copyediting and production can begin. Otherwise, you will be notified when your proofs are ready to be viewed.

Sincerely,
Ryan Blaustein
Editor
Microbiology Spectrum